# Lipid Polymorphism of the Subchloroplast—Granum and Stroma Thylakoid Membrane–Particles. II. Structure and Functions

**DOI:** 10.3390/cells10092363

**Published:** 2021-09-09

**Authors:** Ondřej Dlouhý, Václav Karlický, Rameez Arshad, Ottó Zsiros, Ildikó Domonkos, Irena Kurasová, András F. Wacha, Tomas Morosinotto, Attila Bóta, Roman Kouřil, Vladimír Špunda, Győző Garab

**Affiliations:** 1Group of Biophysics, Department of Physics, Faculty of Science, University of Ostrava, 710 00 Ostrava, Czech Republic; ondrej.dlouhy@osu.cz (O.D.); vaclav.karlicky@osu.cz (V.K.); irena.kurasova@osu.cz (I.K.); vladimir.spunda@osu.cz (V.Š.); 2Laboratory of Ecological Plant Physiology, Domain of Environmental Effects on Terrestrial Ecosystems, Global Change Research Institute of the Czech Academy of Sciences, 603 00 Brno, Czech Republic; 3Department of Biophysics, Centre of the Region Haná for Biotechnological and Agricultural Research, Palacký University, 783 71 Olomouc, Czech Republic; r.arshad@rug.nl (R.A.); roman.kouril@upol.cz (R.K.); 4Groningen Biomolecular Sciences and Biotechnology Institute, University of Groningen, 9700 AB Groningen, The Netherlands; 5Photosynthetic Membranes Group, Institute of Plant Biology, Biological Research Centre, Eötvös Loránd Research Network, 6726 Szeged, Hungary; zsiros.otto@brc.hu (O.Z.); domonkos.ildiko@brc.hu (I.D.); 6Institute of Materials and Environmental Chemistry, Eötvös Loránd Research Network, 1117 Budapest, Hungary; wacha.andras@ttk.hu (A.F.W.); bota.attila@ttk.hu (A.B.); 7Department of Biology, University of Padova, 35131 Padova, Italy; tomas.morosinotto@unipd.it

**Keywords:** bilayer, chlorophyll fluorescence, cryo-electron-tomography, electron microscopy, membrane energization, membrane networks, non-bilayer lipid phases, SAXS, violaxanthin de-epoxidase

## Abstract

In Part I, by using ^31^P-NMR spectroscopy, we have shown that isolated granum and stroma thylakoid membranes (TMs), in addition to the bilayer, display two isotropic phases and an inverted hexagonal (H_II_) phase; saturation transfer experiments and selective effects of lipase and thermal treatments have shown that these phases arise from distinct, yet interconnectable structural entities. To obtain information on the functional roles and origin of the different lipid phases, here we performed spectroscopic measurements and inspected the ultrastructure of these TM fragments. Circular dichroism, 77 K fluorescence emission spectroscopy, and variable chlorophyll-a fluorescence measurements revealed only minor lipase- or thermally induced changes in the photosynthetic machinery. Electrochromic absorbance transients showed that the TM fragments were re-sealed, and the vesicles largely retained their impermeabilities after lipase treatments—in line with the low susceptibility of the bilayer against the same treatment, as reflected by our ^31^P-NMR spectroscopy. Signatures of H_II_-phase could not be discerned with small-angle X-ray scattering—but traces of H_II_ structures, without long-range order, were found by freeze-fracture electron microscopy (FF-EM) and cryo-electron tomography (CET). EM and CET images also revealed the presence of small vesicles and fusion of membrane particles, which might account for one of the isotropic phases. Interaction of VDE (violaxanthin de-epoxidase, detected by Western blot technique in both membrane fragments) with TM lipids might account for the other isotropic phase. In general, non-bilayer lipids are proposed to play role in the self-assembly of the highly organized yet dynamic TM network in chloroplasts.

## 1. Introduction

In the accompanying paper (Part I) in the same issue, we have shown that the granum and stroma thylakoid membrane (TM) sub-chloroplast particles, obtained by digitonin fragmentation of spinach TMs, exhibit marked lipid polymorphism [1]. Similar to intact TMs, they have been shown to contain an inverted hexagonal (H_II_) phase and two isotropic phases (I_1_ and I_2_) in addition to L, the lamellar (bilayer) phase. It was also shown that thermal and lipase treatments of these membranes exerted well discernible specific effects on the lipid phases: thermal treatments destabilized the bilayer and H_II_ phases and substantially increased the contribution of the I_1_ phase; this latter phase was highly susceptible to wheat germ lipase, which, at the same time, hardly affected the L and H_II_ phases [1].

The L phase is crucial for the build-up of Δμ_H_^+^ [2], yet about half of the total lipid content of TMs is constituted by monogalactosyldiacylglycerol (MGDG)—a non-bilayer lipid species [3], and TMs contain non-bilayer lipid phases [4,5]. Earlier, we have shown the modulation of TM permeability by isotropic phases [6] and that reversible pH- and temperature-induced changes in the lipid polymorphism were associated with the activity of violaxanthin de-epoxidase (VDE) [7]. In general, non-bilayer lipid phases have been shown to contribute significantly to the structural and functional dynamics of TMs. However, the molecular assemblies responsible for the non-bilayer phases of TM have not been identified.

Here, to shed more light on the nature of the non-bilayer lipid phases and their physiological significance, we carried out spectroscopic experiments, providing information on the organization and functional activities of the photosynthetic apparati, and on the effects of thermal and lipase treatments on granum and stroma TMs. In particular, we performed circular dichroism (CD) and 77 K fluorescence spectroscopy measurements, to obtain information on the molecular organization of the pigment-protein complexes (PPCs) [8] and the distribution of the excitation energy in the membranes [9], respectively. The photochemical activity of Photosystem II, essentially absent in the stroma TMs, was monitored by recording fast chlorophyll-a (Chl *a*) fluorescence transients [10,11,12]. The ability of the granum and stroma TMs to generate and maintain an energized state was determined using the electrochromic absorbance transients (ΔA_515_) [13]. Our data revealed only minor thermal- or lipase-treatment induced alterations in the molecular architecture and functions of the photosynthetic machineries of the two membrane fractions. Further, as revealed by ΔA_515_ measurements, both fragments of the TM system were re-sealed and formed closed vesicles, which largely retained the impermeability of their membranes after lipase treatments.

We also investigated the ultrastructure of these membrane preparations to identify possible structural entities that can be held responsible for the presence of the different non-bilayer lipid phases. Small-angle X-ray scattering (SAXS) was measured to characterize the membranes and the multilamellar organization of grana, and to attempt detecting the H_II_ phases. SAXS is a powerful technique, which carries information on membrane systems on the mesoscopic scale of 1–100 nm [14,15]. The ultrastructure of granum and stroma TMs were inspected with the aid of freeze-fracture and scanning EM techniques, FF-EM and SEM, respectively, and by cryo-electron tomography (CET). These techniques have provided a wealth of information about the organization of supercomplexes, lipid phases, and the ultrastructure of TMs [16,17,18,19,20,21], while not detected by SAXS, FF-EM and CET images were found to contain membrane-associated structures with features of the H_II_ phase. Further, EM and CET data revealed extensive spontaneous formations of membrane networks, which is thought to be relevant to the self-assembly of TM networks *in vivo.* Based on the presented data, we suggest, in agreement with our previous hypotheses [5], that one of the isotropic phases originates from fusion channels—structures responsible for interconnecting protein-rich domains and fusing originally isolated membrane fragments; and that the other isotropic phase arises from VDE-lipid interactions—the presence of this water-soluble lipocalin-like enzyme was detected in both membrane particles.

## 2. Materials and Methods

### 2.1. Isolation of Granum and Stroma Thylakoid Membranes

The granum and stroma subchloroplast thylakoid membrane particles were isolated by digitonin fragmentation of spinach TMs, followed by differential centrifugation, using modified protocols [22,23], as described in Part I [1]. This isolation procedure has been shown to largely retain the in vivo properties of the granum and stroma thylakoids [24].

### 2.2. Lipase Treatments and Measurements at Different Temperatures

Wheat germ lipase (L3001, Sigma-Aldrich, Burlington, MA, USA) stock solution of 0.5 U µL^−1^ activity was prepared in Milli-Q water. Lipase treatments of granum and stroma TMs were performed under conditions similar to those applied in the ^31^P-NMR experiments, at 5 °C in the dark [1]. Briefly, the concentrated suspension of membrane fractions (Chl concentration of 10 mg mL^−1^) was incubated for 1 h before the control (0 U) measurement was carried out, then the required amount of lipase was added, and each consecutive lipase-treated sample was incubated for 1 h before the measurement.

For the measurements at different temperatures, the concentrated sample was placed into CH-100 heating/cooling dry block (Biosan, Rīga, Latvia) and incubated for 1 h at 5 °C; then the temperature was raised to 15 °C and from 15 to 25 °C after another 1 h incubation. At the desired temperatures, aliquots were taken for the preparation of diluted samples. Electrochromic absorbance transients were recorded at 5 °C.

### 2.3. Circular Dichroism Spectroscopy

Circular dichroism (CD) spectra of granum and stroma TMs were measured in the range of 350–750 nm with a J-815 spectropolarimeter (Jasco, Tokyo, Japan). The spectra were recorded in steps of 0.5 nm with an integration time of 1 s, a band-pass of 2 nm and scanning speed of 100 nm min^-1^ in a cell with an optical path length of 1 cm at Chl concentration of 20 µg mL^−1^. The temperature of the samples was controlled by Peltier sample holder (Jasco, Tokyo, Japan) and set at 5, 15, and or 25 °C. The CD spectra were normalized to the Chl Q_y_ absorption band. 

### 2.4. 77 K Chl a Fluorescence Spectroscopy

Chl *a* fluorescence spectra of granum and stroma TMs at 77 K were recorded using a Fluorolog 3–22 spectrofluorometer (Horiba Jobin Yvon, Paris, France) equipped with a Dewar-type optical cryostat. Fluorescence spectra were measured on diluted suspension of granum and stroma TMs in a 2 mm capillary tube; the Chl content was adjusted to 5 µg mL^−1^ to avoid reabsorption. The emission spectra were recorded at the preferential excitation of Chl *a* at 435 nm with 3 and 2 nm slit widths of excitation and emission monochromators, respectively. The emission spectra were corrected for the spectral sensitivity of the detection system and normalized at 685 and 730 nm for granum and stroma TMs, respectively.

### 2.5. Fast Chl a Fluorescence Transients

Dark-adapted samples at a Chl content of 50 µg mL^−1^ were measured in a cuvette with an optical path length of 1 cm, with a FluorPen FP 100 fluorometer (Photon Systems Instruments, Drásov, Czech Republic), employing blue LEDs of 3000 µmol photons m^−2^ s^−1^ for excitation. The *F_v_/F_m_* ratio, the maximal photochemical efficiency of PSII, was calculated as (*F_m_*−*F*_0_)/*F_m_*, where *F_0_* and *F_m_* are minimal and maximal fluorescence levels.

### 2.6. Electrochromic Absorbance Transients, ΔA_515_

The electrochromic shift (ECS) or ΔA_515_ of dark-adapted samples with a Chl content of 20 µg mL^−1^ was measured in a 1 cm × 1 cm cuvette using a Joliot-type kinetic spectrometer (JTS-100, BioLogic, Grenoble, France). The detectors were protected with a BG-39 filter; a 520 nm interference filter was placed in front of the pulsed LED source of the measuring flashes. An LF1 xenon-flash light source (Hamamatsu Photonics, Hamamatsu, Japan) was used to activate the sample with a single-turnover saturating flash, applied at 90° with respect to the measuring beam. Five kinetic traces were averaged, with a repetition rate of 0.15 s^−1^. For granum TMs, DAD (2,3,5,6-tetramethyl-1,4-phenylenediamine, VWR International, Radnor, PA, USA) of 100 µM final concentration and FeCy (potassium ferricyanide, Sigma-Aldrich) of 25 µM final concentration were added as a substitute for electron transport chain components lost during the sample isolation; for stroma TMs, phenazine methosulfate (PMS, Sigma-Aldrich) of 100 µM final concentration and sodium ascorbate of 1 mM final concentration were added. Gramicidin (Sigma-Aldrich), a transmembrane ion channel, was dissolved in ethanol (p.a., Mach Chemikálie, Ostrava, Czech Republic) and was added to the sample in a final concentration of 0.25 µM to determine the non-electrochromic component of the signal. Final concentration of ethanol did not exceed 1%.

### 2.7. Data Analysis

Spectroscopic data and kinetics were processed in MATLAB (MathWorks, Natick, MA, USA) using the Spectr-O-Matic toolbox (Dr. Petar H. Lambrev, Szeged, Hungary), which is available at the MATLAB File Exchange and homebuilt routines.

Statistical analysis of *F_v_/F_m_* ratio of lipase- and thermal-treated granum TMs was performed using one-factor ANOVA followed by post-hoc Tukey’s test with significance level at *p* < 0.05. All testing was performed using Origin 8.6 (OriginLab, Northampton, MA, USA).

### 2.8. SDS-PAGE and Western Blotting

Proteins were extracted from TM preparations by dilution in a buffer according to [25] and incubated for 30 min at 70 °C. SDS-PAGE was carried out using 8.5% focusing and 12% resolving polyacrylamide gel containing 0.1 % sodium dodecyl sulfate (SDS, Sigma-Aldrich). The loaded intact TM suspension and the isolated subchloroplast particles corresponded to 2 and 20 µg of total Chl contents, respectively. Proteins were subsequently electrophoresed for 90 min at 120 V. After the electrophoresis, gels were equilibrated in Towbin buffer for 15 min and concurrently PVDF membranes (Bio-Rad Laboratories, Hercules, CA, USA) were pre-wetted in methanol. The proteins were transferred on membrane for 60 min at 120 V in the Towbin buffer using Trans-Blot Cell (Bio-Rad Laboratories, Hercules) cooled at 4 °C. The blotted membranes were washed in TBS buffer (pH 7.6) for 2 × 5 min, followed by overnight incubation with a blocking solution (TBST + 5% milk powder *w*/*v*; Serva Electrophoresis GmbH, Heidelberg, Germany) at 4 °C. After blocking, the membranes were washed with the TBST buffer (2 × 10 min) and incubated for 1 h with anti-VDE antibody (1:3000, AS15 3091, Agrisera, Vännäs, Sweden). The washed membranes (TBST; 2 × 10 min) were incubated for 1 h with the secondary antibody (HRP, 1:30,000, AS09 602, Agrisera, Vännäs, Sweden) and again washed (2 × 5 min TBST; 2 × 5 min in TBS). Blots were visualized using chemiluminescent substrate (#32209, Pierce ECL Western Blotting substrate, Thermo Fisher Scientific, Waltham, MA, USA); chemiluminescence was scanned on ChemiDoc MP gel imager (Bio-Rad Laboratories, Hercules). The molecular weight of detected bands was assigned using Bio-Rad low-range molecular weight protein standard loaded on the gel.

### 2.9. VDE Protein Expression

Mature VDE from *Arabidopsis thaliana* was expressed in *Escherichia coli* Origami B strain after induction with 1 mM isopropyl β-D-1-thiogalactopyranoside (IPTG) for 5 h at 37 °C. VDE was then purified on a nickel affinity chromatography (from Sigma-Aldrich) and eluted in 50 mM HEPES pH 7.5, 50 mM NaCl, 100 mM imidazole, as previously described [26].

### 2.10. Small-Angle X-ray Scattering (SAXS)

Small-angle X-ray scattering measurements were performed using CREDO [27,28], an in-house transmission geometry set-up. Samples were filled into thin-walled quartz capillaries of 1.2 mm average outer diameter. After proper sealing, these were placed in a temperature-controlled aluminum block, which was inserted into the vacuum space of the sample chamber. Measurements were performed using monochromatized and collimated Cu Kα radiation (0.1542 nm wavelength), and the scattering pattern was recorded in the range of 0.02–5 nm^−1^ in terms of the scattering variable, *q* (*q* = (*4π*/*λ*)sin*Θ*), where *2Θ* is the scattering angle and λ is the X-ray wavelength. The total measurement time was 12 h for each sample at three different geometries (three different sample to detector distances to cover the desired scattering interval). Fresh samples were used after changing the geometry. In order to be able to assess sample and instrument stability during the experiment, the exposures were made in 5-min units, with frequent sample change and reference measurements. These individual exposures were corrected for beam flux, geometric effects, sample self-absorption, and instrumental background, as well as calibrated into physical units of momentum transfer (*q*, nm^−1^) and differential scattering cross-section (absolute intensity, cm^−1^ × sr^−1^) [28]. The corrected and calibrated scattering patterns were azimuthally averaged to yield a single one-dimensional scattering curve for the samples.

### 2.11. Freeze-Fracture Electron Microscopy (FF-EM)

Approximately 1–2 μL of sample was pipetted into the hole of a cylinder-shaped gold sample holder and frozen by plunging it immediately into partially solidified Freon for 20 s, and stored in liquid nitrogen until fracturing. Fracturing was performed at 173 K in a freeze-fracture device (BAF 400D, Balzers AG, Balzers, Liechtenstein). The fractured surfaces were etched for 30 s at 173 K then shadowed by platinum and covered with carbon [29]. The replica was washed with surfactant solution and distilled water, and it was transferred to 200 mesh copper grid for transmission electron microscopic (MORGAGNI 268D, FEI, Hillsboro, OR, USA) examination. The resolution of the FF-TEM images is approximately 2 ± 1 nm, due to the average size of the platinum particles formed during the sputtering process.

### 2.12. Scanning Electron Microscopy (SEM)

The membrane fractions were fixed in 2.5% glutaraldehyde for 2 h, settled for 45 min and then filtered on poly-L-lysine—coated polycarbonate filter. After post-fixation in 1% OsO_4_ for 40 min, the samples were dehydrated in aqueous solutions of increasing ethanol concentrations, critical point dried, covered with 5 nm gold by a Quorum Q150T ES (Quorum Technologies, Lewes, UK) sputter, and observed in a JEOL JSM-7100F/LV scanning electron microscope (JEOL, Tokyo, Japan).

### 2.13. Cryo-Electron Tomography (CET)

Preparation of cryo specimens of granum and stroma TMs for cryo-electron tomography was performed according to [30]. Tomographic single-axis tilt series ranging from −64° to +64° (an increment step 2°) were imaged in a Titan Krios microscope (Thermo Fisher Scientific) equipped with Gatan energy filter and K2 direct electron detector (Gatan, Pleasanton, CA, USA) using SerialEM software [31] at 53,000× magnification with specimen level pixel size of 2.84 Å. The electron dose was set between 50–60 e/Å^2^. Tomograms were reconstructed using EMAN2 software [32] and de-noised by iterative non-linear anisotropic diffusion [33] from IMOD software package [34]. Surface views of the membrane vesicles in reconstructed tomograms were produced using the 3dmod program (a part of the IMOD package).

## 3. Results and Discussion

We have shown in Part I of this study [1]—using ^31^P-NMR spectroscopy—that both the isolated granum and stroma TMs contain four well discernible lipid phases, a lamellar phase, two isotropic phases (I_1_ and I_2_), and an H_II_ phase, which form distinct yet interconnected entities. To obtain information about the origin of these different lipid phases and their possible roles in photosynthetic functions, we characterized the main spectroscopic, functional, and structural parameters of the granum and stroma TMs.

### 3.1. CD Spectroscopy

To characterize the molecular organization of the pigment systems we measured the CD spectra of the two types of TMs (Figure 1). Blue curves show the typical spectra of the untreated granum (Panels a and c) and stroma (Panels b and d) TMs at 5 °C. The spectrum of granum TMs is dominated by the excitonic bands of LHCII, displaying band pairs at (−)653 and (+)665 nm in the red, and (+)484 and (−)473 nm in the Soret region [35,36]. The bands at (+)689 and (+)510 nm, the amplitudes of which varied from batch to batch, are attributed to residual psi-type bands reflecting the remaining long-range order of the pigment PPCs in the granum TM preparations. In purified PSII membranes—BBY [37] and grana patches [38], lacking multilamellar organization—these bands are absent [39].

The CD spectrum of stroma TMs is very similar in character to the isolated PSI-LHCI supercomplex [40,41,42] and proteoliposomes reconstituted with PSI [43]. The spectrum contains the characteristic excitonic bands in the Chl *a* Q_y_ region around (+)670 nm and (-)695 nm and a positive carotenoid band around 510 nm. The CD spectrum of granum TMs resembles those of PSII supercomplexes, and PSII-enriched membranes [39,44,45], with some contributions of a residual psi-type band at (+)690. As shown in Figure 1, neither the thermal nor the lipase treatment induced sizeable changes in the band structures of either the granum or the stroma TMs.

The excitonic CD signal is a highly sensitive marker of the molecular organization of the PPCs. For instance, the spectrum of LHCII varies upon minor changes in its physicochemical environment [45]. Hence, the invariance of the excitonic band structure in the granum and stroma TMs indicates that these treatments do not induce significant changes in the molecular organization of PPCs.

### 3.2. 77 K Chl a Fluorescence Spectroscopy

The fluorescence emission signal of Chl *a* in TMs at 77 K originates from LHCII (680 nm), core antennas of PSII (685 and 695 nm), PSI core (720 nm), and LHCI (735 nm) [9]. Figure 2 shows the 77 K fluorescence emission spectra of granum (Panel a) and stroma (Panel b) TMs (blue curves). Granum TMs exhibit intense peaks at 683 and 693 nm, showing the significant emissions from PSII; and at 730 nm, which can be attributed to PSI emission originating from the end membranes of grana. In contrast, the spectrum of stroma TMs is dominated by the PSI band at 730 nm, with weaker bands below 700 nm, probably arising from either LHCII-PSII or LHCI. Our 77 K emission spectra of stroma TMs are closely reminiscent of spectra published earlier on similar preparations [42,46]. As shown in Figure 2, lipase treatments (red curves) induced virtually no change in the spectra. The effect of high temperature on stroma TMs was examined by [42] reporting only minor changes up to 35 °C; elevated temperatures (above 50 °C) caused gradual disassembly of the PSI supercomplex and the denaturation of the antenna complexes. No sizeable changes were seen between the 5 and 25 °C in the 77 K fluorescence emission spectra of granum TMs (data not shown).

In general, the 77 K fluorescence emission spectroscopy carries information about the distribution of the excitation energy among the PPCs. The fact that lipase treatments—which essentially eliminated the sharp ^31^P-NMR isotropic bands [1]—exerted virtually no effect on the emission spectra shows that this treatment does not affect significantly the organization of the supercomplexes embedded in the bilayer membrane. Conversely, the invariance of the 77 K emission bands (and the CD spectra) is in perfect agreement with the observation that the applied lipase treatment had very little effect on the bilayer phase. Hence, these data hint that the isotropic lipid phases are to be found outside the protein-rich region of the TMs.

### 3.3. Fast Chl a Fluorescence Transients

To monitor the functional activity of PSII, we performed measurements of fast Chl *a* fluorescence transients on TM fragments [10,12]. In contrast to grana, the stroma TMs exhibited no sizeable variable fluorescence signal, which allowed us to conclude that there was a very low proportion of PSII, if any, in the stroma TMs. Table 1 shows the *F_v_/F_m_* parameter of lipase- and thermal-treated granum TMs. At 5 °C, the *F_v_/F_m_* value of the control samples varied around 0.6. Elevating the temperature to 25 °C induced a small, albeit statistically insignificant decrease of *F_v_/F_m_* ratio from 0.61 ± 0.07 to 0.46 ± 0.15, evidently due to a destabilization of PSII at a higher temperature. The thermal instability of PSII might, in part, be due to the effect of digitonin; the physicochemical environment of PSII in TMs has been shown to affect its stability [47,48]. In contrast, the lipase treatment did not induce a noticeable change in the functional activity of PSII.

In general, these data show that the functional state of PSII is retained in the granum TMs. More importantly, the fact that PSII activity is not affected by the lipase treatment eliminating the sharp isotropic ^31^P-NMR peaks, strongly suggests that these isotropic signals do not arise from membrane domains containing the LHCII-PSII supercomplexes, but rather originate from regions participating in membrane fusion and/or arise from interactions of lipids with water-soluble proteins, such as VDE and other lipocalins (cf. [5]).

### 3.4. Electrochromic Absorbance Transients, ΔA_515_

To obtain information about the electrogenic activity of the two photosystems, the formation of vesicles and the variations in the membrane permeability, we measured electrochromic absorbance transients at 520 nm (‘ΔA_515_’). The transmembrane electric potential in TMs is given rise by the uniform (delocalized) electric fields generated equally by PSII and PSI primary charge separations and secondary electron transfer steps; ΔA_515_ or ECS is defined as the gramicidin-sensitive absorbance transients [13]. Under repetitive excitation of the samples with single-turnover saturating flashes, ΔA_515_ requires electron donors and acceptors, which are replenished by DAD and FeCy (as PSII electron acceptors, added to the granum TMs) and PMS and sodium ascorbate (as PSI electron donor, added to the stroma TMs).

We used this technique of ECS to test if the dramatic effect of the lipase treatments on the granum and stroma TMs, as reflected by the elimination of the sharp isotropic ^31^P-NMR signal [1], has consequences on the permeability of membranes (Table 2). It can be seen that the ECS amplitudes are only marginally affected.

In general, these data show that after the fragmentation of the TMs by digitonin, i.e., after the splitting of the stroma membranes from the grana, the obtained membrane particles are re-sealed. In other terms, the membrane fragments regain their ability to generate and hold a transmembrane uniform electric field. In intact thylakoids, the granum and stroma TMs enclose a common lumenal space [49]; thus, upon disrupting their junctions, at the sites of fragmentation, the lumen is exposed to the outer aqueous phase. Without re-sealing, it would lead to the loss of the transmembrane electric potential difference. The vesicle formation of stroma TMs, obtained after digitonin fragmentation and differential centrifugation [23], was earlier reported, showing electric potential generation in well -coupled PSI-enriched vesicles [50]. Our data demonstrate the same re-sealing ability of the isolated granum TMs (Table 2).

Regarding the origin of I_1_ and I_2_, detected by ^31^P-NMR spectroscopy (Part I), it is to be stressed that the same lipase treatments which eliminate the sharp isotropic ^31^P-NMR peaks (Part I) exert only minor effects on ΔA_515_. This corroborates the notion that the isotropic phases are to be found outside the bilayer regions of the granum and stroma TM preparations, which agrees well with the proposed membrane model accounting for the isotropic phases [5].

### 3.5. SDS-PAGE and Western Blotting

The activity of VDE is strictly dependent on the presence of non-bilayer phases [51,52] and VDE is capable of binding the non-bilayer lipid MGDG [53]. Earlier we assigned one of the isotropic phases to its association with lipid molecules (cf. [5]). Since the isolated granum and stroma TMs contained significant amounts of isotropic lipid phases, we wanted to test if VDE is still present in these particles. The lengthy isolation procedure—during which the intact thylakoid vesicles were ruptured and, inevitably, the lumenal and stromal aqueous phases were temporarily interconnected—could have easily led to the loss of this lumenal protein. To qualitatively test the presence of VDE in the granum and stroma TM particles, we used SDS-PAGE, for protein separation, and a subsequent Western-blotting, for its immunodetection (Figure 3). The bands containing VDE were assigned using the recombinant VDE protein as a positive control (first column). The duplicity of VDE bands, when using SDS-PAGE under non-reducing conditions, probably originates from the different redox environments of VDE [54]. Alternatively, it might be caused by the presence of a VDE propeptide at ca 52 kDa and of the mature protein, found at ca 40 kDa (according to Agrisera, the antibody manufacturer). In either case, the presence of VDE, albeit at relatively low concentrations, can be detected in both the granum and the stroma TMs. By considering the 10 times higher concentration of the loaded Chl content of the membrane particles compared to intact TMs, the estimated VDE contents in the granum and stroma TMs (third and fourth column, respectively) were approximately 10 times lower compared to the intact TMs (second column). This is not surprising with regards to the digestion of the TMs by digitonin, and further steps of washing during the isolation procedure of the particles.

These data, the presence of VDE in the granum and stroma TM fragments supports the tentative assignment of one of the isotropic ^31^P-NMR peaks, probably the weaker, I_2_ peak as lipids surrounding the VDE protein. In broad terms, this tentative assignment is in agreement with experimental data showing the requirement of non-bilayer lipid phases to activate this lumenal enzyme—both in model membranes [51] and in intact TMs [7]. It also satisfies the condition of this lipid phase not to be included in the bilayer membrane, yet being connected with it. The identity of the I_2_ peak and the lipid phase induced by the interaction of VDE with lipids of non-bilayer propensity should be confirmed by further experiments.

### 3.6. Small Angle X-ray Scattering (SAXS)

To obtain information on the structure of the membrane particles and the presence of H_II_ phase [55], we performed small angle X-ray scattering (SAXS) experiments on the granum and stroma TMs. As it can be observed in Figure 4, these TM particles displayed significantly different SAXS patterns. The distinct shape of curves, especially in the middle range of the scattering variable from 0.7 to 2 nm^−1^, indicates different layer arrangements. The SAXS of grana shows two diffuse reflections at q = 0.95 and 1.58 nm^−1^, corresponding to 6.6 and 4.0 nm, respectively. These values are close to that which were observed in the computed scattering curves of a model grana stack [56]. In our case, the diffraction profiles are smoothed corresponding to a weakly correlated multi-lamellar system, with a repeat distance of ~31 nm. This value agrees reasonably well with that found in intact TMs suspended in sorbitol-based medium—using small-angle neutron scattering [57]. The SAXS curve of stroma TMs exhibits a hump, as an extremely broadened diffraction profile reflecting a drastically loose hierarchy of the stroma lamellae. Additionally, a shoulder appears in the scattering at q ~0.5 nm^−1^. This local scattering increment might be the consequence of the scattering of individual objects, most probably the ATP synthase, the size of which contributes to scattering in the same interval [58]. It must also be pointed out that our SAXS data revealed no indication of the presence of the H_II_ phase.

In general, these data confirm the bilayer organization of the membrane particles, in perfect agreement with the ^31^P-NMR spectroscopy data on the same samples (Part I, [1]). The lack of H_II_ phase in the SAXS profiles, despite the clear and strong ^31^P-NMR signatures, suggests that the lipid molecules constituting the H_II_ phase do not adopt a sufficiently long-range order and are likely to be found in small patches. It is to be noted that by using FF-EM, H_II_ phase has been observed in thylakoid membranes isolated from low light (LL) grown spinach leaves (but not from ordinary-light grown plants) [59]. However, to our knowledge (i) no SAXS data are available under comparable conditions; (ii) no quantitative data are provided on the proportion of lipids found in H_II_ phase in LL TMs—hence the occurrence of such patches may still be undetectable by SAXS, which also carries information on the lamellar order of TMs; (iii) while the H_II_ signature of purified MGDG and the mixture of MGDG and MGDG:LHCII assemblies can be detected by SAXS [60], no such signal could be revealed in spinach TMs under different experimental conditions [56] (see also [61]), despite the strong ^31^P-NMR signatures in TMs under comparable conditions [5,7]. The absence of a well discernible SAXS signature of the H_II_ phase in whole TMs is also most probably due to the lack of long-range order of the lipid arrays assuming this phase, which may broaden the bands, and/or the strong overlap between the relatively weak H_II_ peaks with the much higher intensity and relatively broad lamellar order Bragg peaks of the TMs.

### 3.7. Freeze-Fracture Electron Microscopy (FF-EM)

To characterize the ultrastructure of the granum and stroma TM particles we used FF-EM, which is suitable to identify protein-rich membrane regions [62]. It has also been used to detect the presence of H_II_ phase after co-solute treatment [17,63], long storage of membranes at 5 °C [16], and in TMs isolated from low-light grown spinach [59]. Figure 5 shows FF-EM images of isolated granum and stroma TM particles (Panels a and b, and c and d, respectively).

Stacks of closely packed membranes, corresponding to thylakoid distances, can be observed in the electron micrographs of granum TM particles, which appear to be organized in large networks comprised primarily of bilayers (Figure 5a). Between the granum membrane vesicles, the lumen is also visible. In general, the periodic order of the lamellae is weak, compared to intact chloroplasts (see e.g., [62]), and cannot be observed in all regions—explaining the weak, broad small-angle X-ray reflections (c.f. Figure 4). The protein complexes of granum are visible as protrusions in the face of sheets or dispersed PPCs, which are embedded in the membrane lipid bilayers—as it can be recognized in the inset of Figure 5a. These protein complexes display a relatively narrow size-range, extending from 6 to approximately 12 nm. Besides this surface morphology, we frequently observed loose, less correlated parts, where the structural units are separated into three kinds of domains: the protein-rich region (P), aqueous domains (W) and elongated, rod-shaped, non-lamellar assemblies (NL) (Figure 5b). Grains of different sizes and shapes, composed of tightly packed arrays of the protein complexes (P), and elongated structures (NL) are observed. Among them, small pools with entirely smooth surfaces, presumably aqueous domains (W), appear. The elongated domains do not contain protein particles, their morphology differs from the bilayer (Figure 5a) and they resemble the tubular domains of the H_II_ phase as observed in artificial lipid assemblies [64]; thus, they may represent the amorphous form of H_II_ phase (NL). As the phase-boundary of H_II_ domains is hydrophobic, their supposed local formation must induce aqueous domains corresponding to a hydrophilic-hydrophobic phase-separation. Indeed, small pools of water (W) can be clearly observed between the elongated non-lamellar domains and the densely packed arrays of protein complexes. As a consequence of the lipid-governed self-assemblies, the protein-rich domains (P) of the membranes are separated from the H_II_ lipid phase.

The FF-EM images of the stroma TM particles (Figure 5c,d) show significantly different surface morphology, which is evidently the consequence of its different protein content compared to grana. The structural units of stroma TMs also form a loose extended network system. The fractioned surface indicates an aqueous matrix in a great part, while the faces of stroma membranes or their cross-sections can also be seen (Figure 5c). Under higher magnification (Figure 5d inset), we can observe larger supercomplexes extending from approximately 7 to 20 nm, most likely corresponding to PSI and the ATP synthase, respectively [58,65,66]. In addition, thin and extremely elongated forms appear to interconnect the protein-rich membrane domains. These elongated assemblies might originate from cross-sectional fractures of single membranes, but they might, in part, be comprised of non-bilayer formations.

In general, these data clearly show the ability of the isolated subchloroplast TM fragments to assemble into large, extended membrane networks. In addition, some regions of these particles appear to indicate the presence of non-bilayer lipid formations—but without well-defined long-range order of the molecular assemblies.

### 3.8. Scanning Electron Microscopy (SEM)

SEM images of granum and stroma TMs (Figure 6, Panels a and b, respectively) confirm the fusogenic ability of these membrane particles, as also demonstrated by FF-EM. Note that the granum and stroma TMs, with typical diameters in the range of 300–500 nm and about 100 nm, respectively, were isolated after digitonin fragmentation of the TM, followed by differential centrifugation at 10,000 and 130,000× *g*. Nevertheless, as testified by SEM (and FF-EM), they readily assemble into large membrane networks. In addition, both granum and stroma membrane particles appear to contain and be in close connection with small lipid vesicles. These small vesicles might have been produced upon the partial solubilization of the TMs by digitonin; i.e., they might not be present in intact TM preparations. Nevertheless, these small vesicles of 30–50 nm diameter might, at least in part, be responsible for one of the isotropic ^31^P-NMR signals in the membrane fragments (Part I) (cf. [67]).

### 3.9. Cryo-Electron Tomography (CET)

To substantiate our conclusions on the fusion of isolated membrane fragments and the formation of small vesicles, and to attempt detecting membrane-associated lipid assemblies with H_II_ geometry, we performed cryo-electron tomography. Figure 7a,b show the 3D model of multilamellar granum TMs. Evidently, the TMs of two adjacent grana partially are fused, as observable in more detail in the tomographic reconstruction (Appendix A). The tomographic reconstruction also revealed characteristic strong densities of headpieces of ATP synthase at peripheral membranes and areas with symmetrical densities of rectangular shape characteristic of PSII core complexes (Appendix A) (for a comparison see [66,68]). We also identified small lipid vesicles (Figure 7a,b, yellow), which are outside the granum TM with either partial or no contact with the membrane. In addition, two lipid vesicles spanning the membrane vesicle can be identified (Figure 7a,b, red), which might participate in lipid fusion between two adjacent membranes [69].

Figure 7c,d show the 3D model of the stroma TMs as they are fused together to assemble into large vesicular structures. The tomographic reconstruction revealed highly abundant ATP synthase in the stromal membranes and smaller densities likely representing PSI complexes (Appendix A). Moreover, in the stroma TM, a small lipid vesicle was observed inside the membrane (Figure 7b,c, in red), which can represent the same type of the fusion channel between the adjacent membranes as we observed in the granum TM. Moreover, a macro-structure composed of stacked small tubular-shaped assembly was revealed in the stroma membrane, supposedly composed of lipids in the H_II_ phase (Figure 7c,d, in salmon); they do not appear to form long-range order of the constituents. We would like to note here that CET of granum and stroma TMs, unlike in intact TMs [20,66], do not show the presence of plastoglobuli. Nevertheless, contributions from TM-attached plastoglobuli to the polymorphism and dynamics of TM lipids cannot be ruled out [70].

In general, our CET images of granum and stroma TMs confirm the presence of small vesicles and the fusion of membranes. Small vesicles and the fused membrane regions can be held responsible for at least one of the two isotropic phases [67,71], most probably the I_1_ phase. Non-bilayer lipids are known to play a role in forming intermediate structures that are involved in membrane junctions and fusion [71,72,73,74]. In particular, in plant chloroplasts, the granum-stroma junctions [75,76], bifurcation of membranes, and the junction of the right- and left-handed helices [20] are difficult to envision without the involvement of non-bilayer phases. Further, the observed membrane associated tubular structures, albeit without apparent long-range order, might give rise to the H_II_ phase detected by ^31^P-NMR spectroscopy. Hence, these findings qualitatively offer an explanation for the observed distinct, yet interconnected lipid phases.

## 4. Conclusions and Perspectives

In this study, we characterized the spectroscopic, functional, and structural properties of isolated granum and stroma TMs. It was found that protein interactions with bulk lipids appeared not to determine the polymorphism of TM lipids. On the other hand, *in vitro* and *in vivo* data have shown that the activity of the lumenal enzyme VDE, and thus lipid-VDE and possibly other lipid–lipocalin interactions might be associated with non-bilayer lipid phases. We also found that lipid phases appeared not to significantly affect the molecular organization and activity of the photosynthetic machinery—consistent with the tight organization and autonomous functioning of PSII and PSI supercomplexes. Concerning the role of variations in the local pressure [77] and the curvature elastic energy [78], due to the presence of non-bilayer lipids in the bilayer membrane, we propose that rather than regulating the functions of the two photosystems, they play roles in their assembly—in analogy to the role of the non-bilayer lipid cardiolipin, which has been shown to facilitate the supercomplex-formation of the mitochondrial respiratory chain [79,80,81,82].

Concerning the structural roles of non-bilayer lipids in the TMs, the data presented here, the role in the reforming of the well-sealed closed vesicles strongly suggest a role in the self-assembly of the TM system. This step involves fusing the lipid membranes; the fusogenic capability of membranes is also evident from the formation of extended networks. As it can be inferred from physicochemical models of lipid mixtures [71], the fusion of membranes is largely facilitated by the non-bilayer propensity of their bulk lipid molecules. Hence, non-bilayer lipids and non-lamellar lipid phases are proposed to be involved in the spontaneous and dynamic networking of TMs. Although further studies are required to elucidate the structural roles of non-bilayer lipid phases, we can safely conclude that the structural data reported here are consistent with the polymorphic phase behavior of TM lipids revealed by ^31^P-NMR spectroscopy (Part I). In general, non-bilayer lipids, via their fusogenic nature and their ability to segregate from and to enter the bilayer membrane, are proposed to contribute significantly to the structural dynamics of TMs—in harmony with the dynamic exchange model (DEM) [5].

With regard to the physiological roles of the non-bilayer lipid phases, we can rely on some firm observations, but ample room is allowed for hypotheses and speculations. Our earlier data have provided irrevocable evidence for the co-existence of and interactions between the bilayer and non-bilayer lipid phases in fully functional isolated plant TMs [4,5]. Moreover, sizable, largely reversible variations in the polymorphic phase behavior of TMs—induced by changing the temperature and the physicochemical environment (pH, osmotic and ionic strengths) of the membranes [4,5,48]—have been documented. Recently, reversible temperature- and low-pH induced enhancements of the isotropic lipid phase(s) of TMs were shown correlated with the enhanced rate of the activity of VDE; the activity of VDE increased despite the acceleration of the decay of the transmembrane electrochemical potential gradient, including the ΔpH [7]. In the light of our data on the structure and function of plant TMs, it appears that the stability of the bilayer and avoiding the formation of non-bilayer lipid phases do not play such a significant role in the energy transduction as it is generally assumed. In particular, all data suggest that the elevation of temperature, from 5 to 15 or 25 °C, largely destabilizes the bilayers and increases the contributions of the non-bilayer lipid phases, parallel with substantial rises in the permeability of the membrane, due to basal ion fluxes [6]. As inferred from the literature data [83,84] and own unpublished measurements, in the physiological temperature interval, parallel with the enhancements of the non-bilayer lipid phases (and increased membrane permeability and fluidity), the rates of electron transport and synthesis of ATP increase. It is unclear if these apparently opposite effects arise merely from a ‘compromise’ between the energy transduction and the structural flexibility of membranes. According to this hypothesis, non-bilayer lipid phases might just lend special attributes to the membranes, which would represent higher value than the disadvantages due to their negative effects on the membrane energization. Special attributes of TMs, which may justify such a compromise, include the operation of some enzymes (e.g., VDE), facilitating the assembly of supercomplexes (see above), membrane fusions, and the self-regulation of the lipid-to-protein ratio, which has been proposed to warrant the high protein to lipid ratio in all energy converting membranes [85]. As an alternative, a non-conflicting hypothesis is that non-bilayer lipids and non-lamellar lipid phases possess more specific functions in the energy conversion. Recently, using fully functional isolated animal inner mitochondrial membranes, strong correlation has been established between the rate of ATP synthesis and the appearance of a non-bilayer lipid phase [86]—suggesting a more direct correlation between the efficiency of the energy conversion and the existence of non-bilayer lipid phases in mitochondria. It is a reasonable hypothesis that similar correlation exists in TMs; this could be tested by measuring the hydrolysis of ATP and the ^31^P-NMR of isolated TMs. In this context, it can be speculated that non-bilayer lipids play direct roles in the lateral proton translocation [87]—as opposed to the formation and utilization of the transversal delocalized proton currents between the two aqueous phases [2]. These questions and, in general, the roles of non-bilayer lipid phases in the self-assembly and in the structural and functional plasticity of TMs, and of other energy converting membranes, require further studies.

## Figures and Tables

**Figure 1 cells-10-02363-f001:**
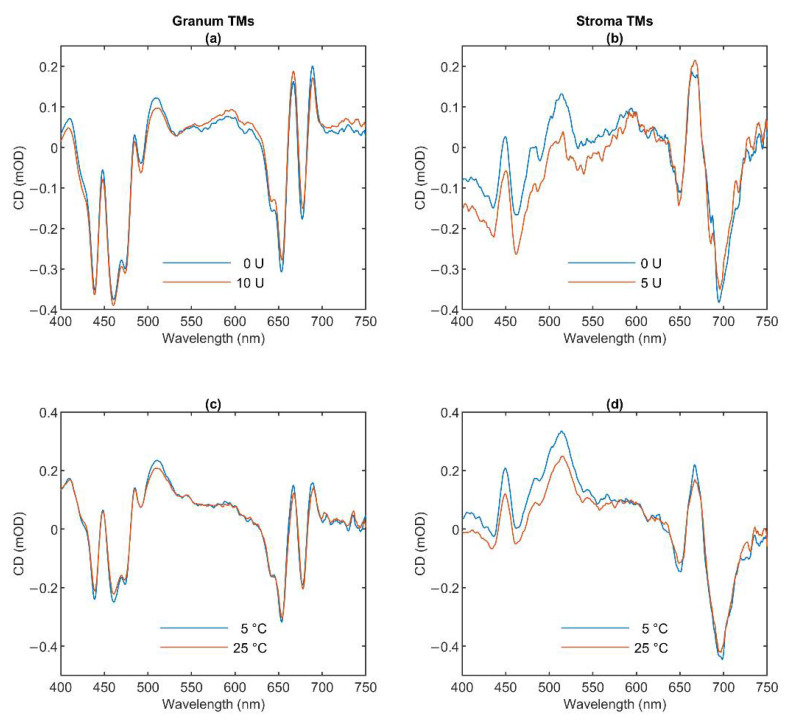
Circular dichroism (CD) of lipase-treated (**a**,**b**) and thermally treated (**c**,**d**) granum (**a**,**c**) and stroma (**b**,**d**) thylakoid membranes (TMs). The activity of the wheat germ lipase (**a**,**b**) was 0 U (blue curves) and 10 U for granum and 5 U for stroma TMs (red curves). The effect of temperature (**c**,**d**) was recorded at 5 °C (blue curves) and 25 °C (red curves). Four individual batches of granum TMs were measured and averaged. We used two different batches of stroma TMs for measuring the effects of thermal- and lipase treatments.

**Figure 2 cells-10-02363-f002:**
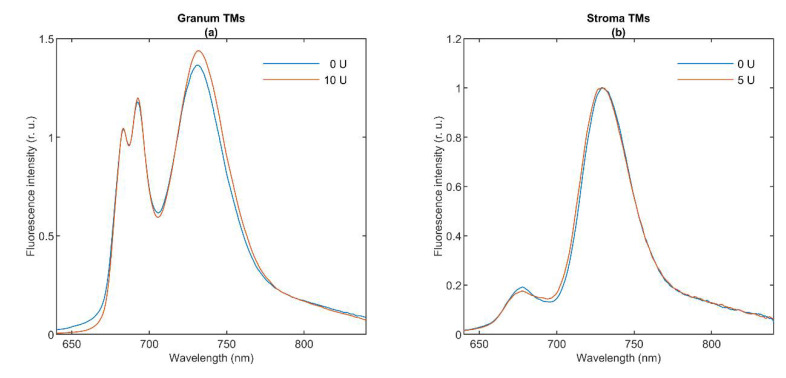
The 77 K fluorescence emission spectra of lipase-treated granum (**a**) and stroma (**b**) thylakoid membranes. Excitation at 435 nm; spectra are normalized at 685 nm ((**a**); emission of PSII) and at 730 nm ((**b**); emission of PSI). All experiments were performed at 5 °C on dark-adapted samples. Four individual batches of granum TMs were measured and averaged. A single batch of stroma TMs was used. (For further details, see Methods.).

**Figure 3 cells-10-02363-f003:**
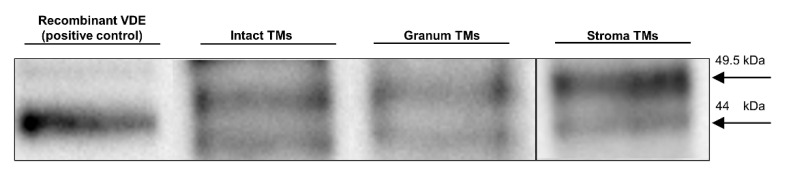
Immunoblotting of a recombinant VDE (positive control, with a MW slightly affected because of the His-tag) and VDE in intact TMs and in granum and stroma membrane particles. Proteins were separated and VDE detected using SDS-PAGE/western blotting. Total Chl contents loaded in the wells were 2 and 20 µg for intact TMs subchloroplast particles, respectively. MW of the bands, 49.5 kDa (upper) and 44.0 kDa (lower). Images of two equally treated blots were processed for granum and stroma TMs. Exposure times: 38 s (intact and granum TMs) and 30 s (stroma TMs).

**Figure 4 cells-10-02363-f004:**
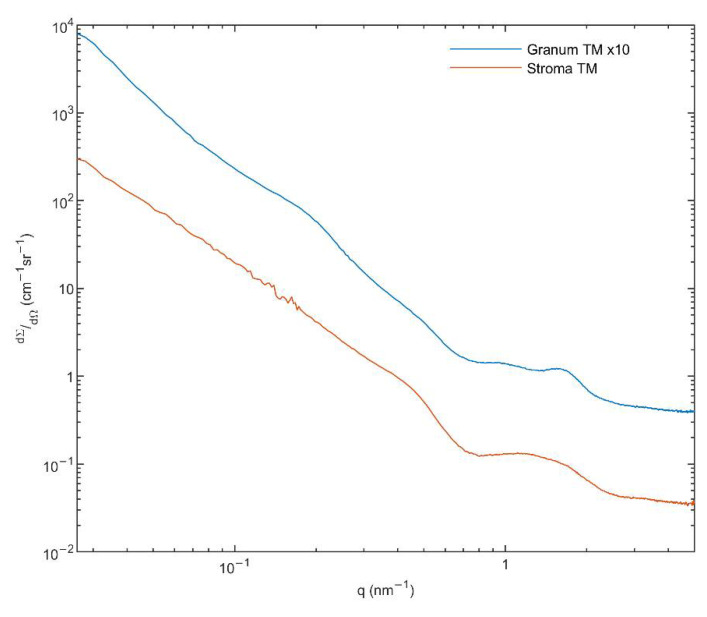
Small angle X-ray scattering (SAXS) of granum (blue) and stroma (red) TMs. Monochromatized and collimated Cu Kα radiation, with a 0.1542 nm wavelength was used. Scattering pattern was recorded in the range of 0.02–5 nm^−1^. Total measurement time was 12 h for each sample at three geometries. For an easier comparison, the SAXS signal of granum TMs was multiplied by a factor of 10. (For further details, see Methods.).

**Figure 5 cells-10-02363-f005:**
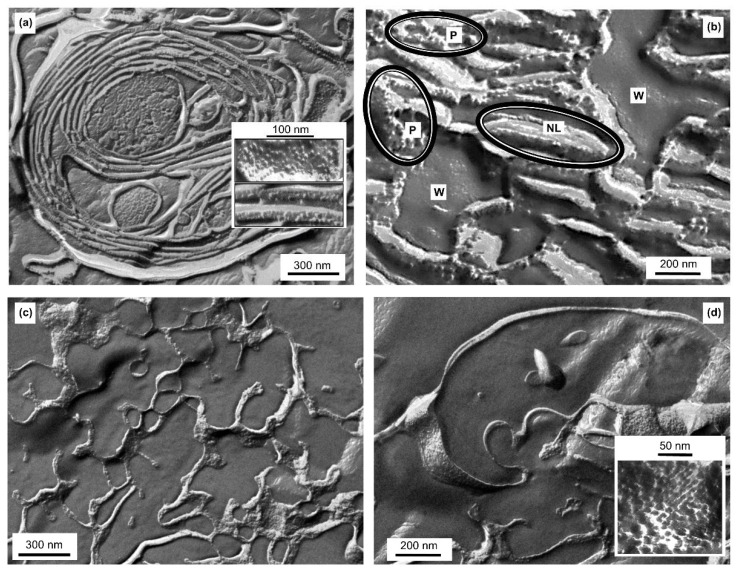
Freeze-fracture electron microscopy images of granum (**a**,**b**) and stroma (**c**,**d**) TMs; images of different regions with different magnifications; insets in (**a**,**d**), protein rich regions; P, W, and NL in (**b**) stand for regions dominated by proteins, water and non-bilayer lipid phase.

**Figure 6 cells-10-02363-f006:**
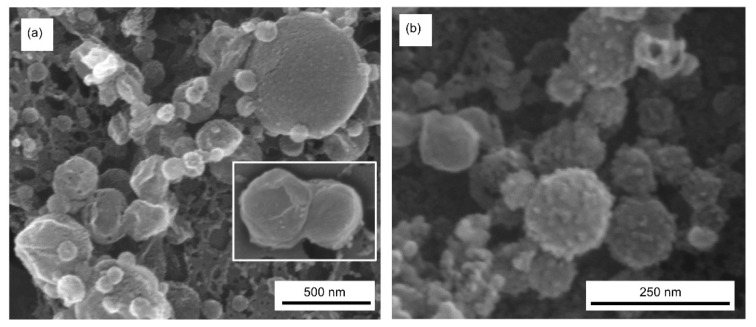
Scanning electron microscopy (SEM) images of granum (**a**) and stroma (**b**) TMs, captured using accelerating voltages of 2 and 15 kV, respectively, of the EM. Inset in (**a**), an example of two apparently fused grana.

**Figure 7 cells-10-02363-f007:**
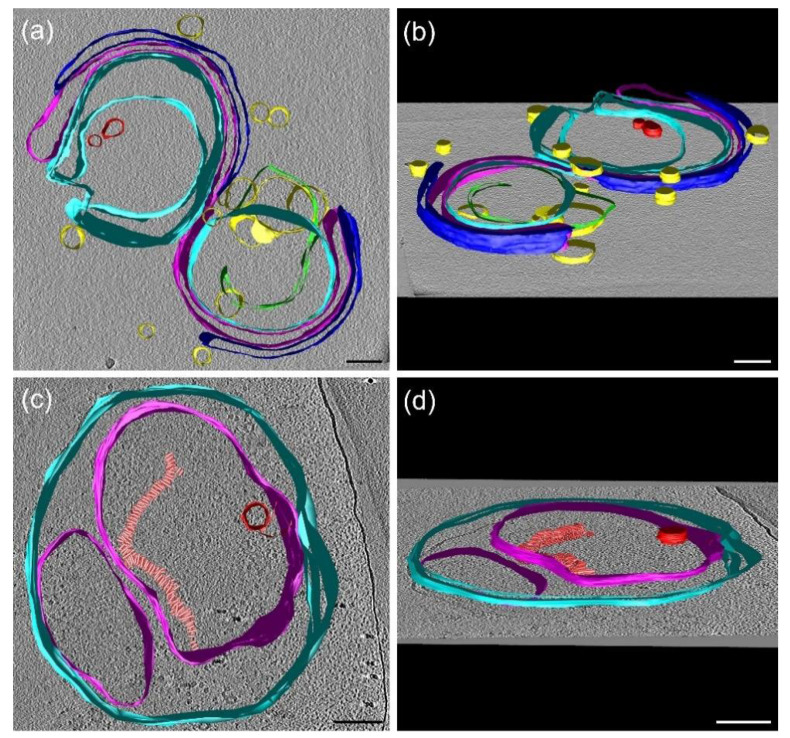
Surface views of reconstructed tomograms of granum and stroma thylakoid membranes. Panels (**a**,**c**) represent the top views of the granum and stroma thylakoid membranes, respectively. Panels (**b**,**d**) represent their corresponding tilted views. Cross-sections of the membranes of individual thylakoids, which are perpendicular to the X-Y image plane, are marked in cyan, magenta, and blue, small membrane vesicles are depicted in yellow. Membrane vesicles, which are inside the thylakoids, are shown in red. A tubular assembly in the stromal thylakoid membrane (**c**,**d**, in salmon) is supposedly formed by lipids in the H_II_ phase. Scale bars, 50 nm.

**Table 1 cells-10-02363-t001:** *F_v_/F_m_* parameter, determined from fast Chl *a* fluorescence induction transient measurements, of granum thylakoid membranes at different, gradually elevated temperatures and after treating the membranes with different activities of wheat germ lipase; errors represent standard deviation from four batches.

Temperature	5 °C	15 °C	25 °C
*F_v_/F_m_*	0.61 ± 0.07	0.59 ± 0.06	0.46 ± 0.15
Lipase activity	0 U	5 U	10 U
*F_v_/F_m_*	0.59 ± 0.05	0.58 ± 0.06	0.57 ± 0.04

**Table 2 cells-10-02363-t002:** Initial amplitudes of the gramicidin-sensitive, electrochromic absorbance transients (ECS) of granum and stroma thylakoid membranes treated with 0 and 10 U, and 0 and 5 U wheat germ lipase, respectively. The ECS transients were measured at 520 nm. Mean values and standard deviations for the granum TMs were obtained from three batches; one batch of stroma TMs was used.

**Granum TMs**	**0 U**	**10 U**
	(1.04 ± 0.20) × 10−^3^	(0.92 ± 0.06) × 10−^3^
**Stroma TMs**	**0 U**	**5 U**
	0.95 × 10^−3^	0.78 × 10^−3^

## Data Availability

Processed and derived data are available from the corresponding author G.G. on request.

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
