# Peer review of "Lipid Polymorphism of the Subchloroplast—Granum and Stroma Thylakoid Membrane–Particles. II. Structure and Functions"

_cells, 2021, doi:10.3390/cells10092363_

Round 1
Reviewer 1 Report
Following Part I, the authors tried to investigate what are the ultrastucture difference between the granum and the stroma lamellae besides the protein composition with diverse techniques. They used CD spectroscopy, fluorescence, electrochromic absorbance to analyze the molecular organization of the pigment system. They confirm that the granum contain mainly LHCII and PSII whereas the stroma lamellae contain PSI-LHCI. They didn’t see any huge influence of the lipase or temperature treatment on these compartimentation. They then try to investigate the VDE localization by SDS-PAGE and Western blot and conclude it is associated to the I2 peak identified in P-NMR experiment of Part I. They continue their analysis by doing SAXS experiment and do not see any difference between the two fractions. Finally they realized freeze-fracture electron microscopy where they see differences between granum fraction and stroma lamellae fraction. They claim that the elongated structures we observed are non-bilayer lipid phase. They finally realize SEM and cryo-electron tomography were they could see vesicles for both fractions. However, the granum fraction contain small vesicles that could be responsible for one of the two isotropic phase.
The author used a wide panel of technic but there is maybe too much data that make the manuscript a bit confusing and hard to follow. A lot of data show what is already known, the segregation PSII-LHCII vs PSI-LHCI in the granum vs the stroma lamellae. The Western blot of VDE does not support any additional information and they use it to claim that VDE belongs probably to the I2 peak…SAXs experiment that might provide a lot of insight is not exploited. The only experiments that seem to provide new differences between granum and stroma lamellae are the microscopy experiments. However, their interpretation try to reconcile P-NMR results that do not indicate differences between fractions.
Major remark:
- The use of digitonin to separate stroma lamellae from granum is a really efficient tool to isolate these fractions in terms of protein segregation. What is the effect of digitonin on membrane lipid? Does it not scramble all the lipids between membranes? The pictures of SEM and CET indicate that membranes reorganize into vesicles. It seems to be pretty far from the in vivo organization. Therefore, I don’t know how the obtained results could be transcribed to in vivo thylakoid organization.
-The Western blot experiment is realized without any loading control. How this figure supports the presence of VDE in the isotropic phase? I do not understand the conclusion of this figure.
- In freeze-fracture experiment, what are the evidence indicating the elongated structure are non bilayer phase? Why these structures are absent from the stroma lamellae fraction whereas hexagonal phases are also very abundant in these fractions?
- The isotropic phases are present in both fractions with a similar contribution according to Part I. Small vesicles are identified in the granum fraction and the authors claim they could be responsible for the isotropic contribution. What are the evidence that allow the attribution of these vesicles to the isotropic phase contribution? And therefore why the small vesicles are observed only in the granum fraction if the isotropic phase is present everywhere?
Reviewer 2 Report
This MS complements nicely the part I (NMR) and as mentioned there, it would be nice to have a more thorough discussion into the driving force (functioning) why those membranes need this kind of behaviour.
Author Response
We are grateful to Reviewer II for carefully reading our manuscript and for his/her helpful comment upon which we extended the section of Conclusions (now, Conclusions and Perspectives).
Comment: “This MS complements nicely the part I (NMR) and as mentioned there, it would be nice to have a more thorough discussion into the driving force (functioning) why those membranes need this kind of behaviour.”
Response: As in our response to the comment of Reviewer II for Part I, we dedicated a paragraph to sum up the literature data and our views on this basic problem of bioenergetics and membrane biology.
We trust that with this addition Reviewer II and the Editor will find our paper suitable for publication in Cells.
Sincerely, Győző Garab on behalf of all co-authors.
Round 2
Reviewer 1 Report
The authors improved their manuscript but there is still some assertive conclusion that are not fully supported by the experiment. The authors might be right in their conclusions but they would need further experiment to prove it. The perspective part that is added at the end of the manuscript is well written and gives interesting insights to investigate.
Major remarks:
- For the SAXS experiment, if it is not more exploited and I agree with the authors, it could be the object of another manuscript, it should be removed. The conclusion of the SAXS result “suggests that the lipid molecules constituting the HII phase do not adopt a sufficiently long-range order and are likely to be found in small patches”. In freeze fracture electronic microscopy, non-bilayer phases could be visible over 200 nm long. It is indeed much smaller that what could be seen in the DPPE–DPPG/water system (over 1 µm long). Is SAXS not sensitive enough to see a 200 nm long HII phase?
- The authors indicate that “small vesicles of 30-50 nm diameter are known to give rise to isotropic signal in 31P-NMR spectroscopy”. Could these vesicles be or derived from plastoglobuli? Could they be the one that are sensitive to the lipase action related to part I results? In intact thylakoid, in their previous paper (Dlouhy et al., 2020), the authors found also isotropic signal whereas plastoglobuli could be seen and “these small vesicles are unlikely to be present in the intact thylakoid membranes”. Why plastoglobuli are not found in digitonin fraction but are present in intact thylakoid fraction? How could you explain the isotropic signal is related to these vesicles if they are absent in intact thylakoid membranes? To clarify these aspects, the microscopy chapters need more explanation.
- For the VDE experiment, I agree with the authors that VDE needs non-bilayer phase to work. However to attribute the isotropic phase to the lipid:lipocalin/VDE assemblies is still speculative and is not demonstrated by this western blot experiment. In the published model (Garab, 2017), it is described as a possibility and not as an assignment as it is written line 367 “Earlier we assigned one of the isotropic phases to its association with lipid molecules”. The western blot indicates only that VDE is bound to membrane either in the granum or stroma lamellae but does not support an assignment to the isotropic phase. The authors might be right about their interpretation but it is not demonstrated by the experiments they provide.
Round 3
Reviewer 1 Report
The authors reply to most of my concerns.